# Analysis and optimization of 15-minute community life circle based on supply and demand matching: A case study of Shanghai

**Haoyuan Wu, Liangxu Wang\*, Zhonghao Zhang, Jun Gao**

School of Environmental & Geographical Sciences, Shanghai Normal University, Shanghai, China

\* wangliangxu@shnu.edu.cn

**Data Availability Statement:** The data underlying the results presented in the study are available from: 1.https://opendata.pku.edu.cn/ 2.https://landscan.ornl.gov/landscan-datasets 3.https://

## Abstract

The 15-minute community life circle (15min-CLC) strategy is one of Shanghai's important methods for building a global city and facing a society with a more diverse population structure in the future. In the existing research, the balance between the construction of the life circle and the needs of the people in the life circle still needs to be further fulfilled. This paper is based on the city's multi-source large data set including 2018 AutoNavi POI (Point of Interests), OSM (OpenStreetMap) road network data and LandScan population data set, and evaluates the current status of Shanghai's 15min-CLC through the fusion of kernel density estimation, service area analysis and other statistical models and proposes relevant optimization suggestions. The results show that there are the following shortcomings: (1) From the perspective of different types of infrastructure service facilities, the spatial construction of Shanghai's overall life service facilities and shopping service facilities needs to be optimized. (2) From the perspective of comprehensive evaluation, the comprehensive service convenience of infrastructure service facilities in the downtown area is relatively high, while the comprehensive service convenience of urban infrastructure service facilities in the suburbs and outer suburbs is relatively low; The diversity of basic service facilities in the 15min-CLC in the downtown area is more consistent with the population distribution; However, in the peripheral areas of the urban area, too many infrastructure service facilities have been constructed. Based on the above shortcomings and the perspective of supply and demand matching, relevant optimization strategies are proposed in different regions and different types of infrastructure service facilities: (1) focus on the construction of basic service facilities in the urban fringe and urban-rural areas, improve the full coverage of the basic service facilities, and appropriately reduce the number of basic service facilities in the downtown area. (2) The development of community business models can be used to promote the development of new life service facilities and shopping service facilities. (3) Improve community medical institutions through facility function conversion, merger and reconstruction, etc. (4) Optimize the hierarchical basic service facility system and improve the population supporting facilities of basic service facilities in the 15min-CLC. This paper incorporates people's needs and concerns on the living environment into the 15min-CLC evaluation model, and uses Shanghai as an example to conduct research, summarizes the existing shortcomings, and proposes corresponding optimization strategies based on the matching

www.openstreetmap.org/#map=9/31.2762/121.4484.

**Funding:** This research was funded by the National Natural Science Foundation of China (No.41730642).

**Competing interests:** The authors have declared that no competing interests exist.

of supply and demand. This article attempts to explore a replicable 15min-CLC planning model, so that it can be extended to the Yangtze River Delta urban agglomeration, to provide reference for further research on the 15min-CLC, and to promote urban construction under the concept of sustainable development.

## Introduction

The rapid urbanization process has had a huge impact on the social, economic, and environmental aspects of the city. People are paying more and more attention to the pursuit of the quality of daily life. Urban research has also gradually shifted from "place-oriented" to "people-oriented", with more emphasis on sustainable development. The report "Big Earth Data in Support of the Sustainable Development Goals (SDGs)" pointed out that open space is a prerequisite for improving the quality of life. SDG 11.7 clearly proposes that by 2030, provide comprehensive, convenient, green public space for all, especially for women, children, the elderly, and those with disabilities [1]. In 2020, we are facing the COVID-19 and living in the community where we live for a long time. In the post-epidemic era, it is particularly important for the sustainable development of the community. Therefore, how to plan public space has become a crucial issue. As the ideological cognition and interest preferences of life groups become more complex and diverse, the difficulty of community construction, operation and governance has increased accordingly. With the continuous transformation of urban planning, the spatial organization model has also changed from the previous residential space organization model of "residential area-residential quarter-residential group" to a hierarchical model of "15-minute living circle-ten-minute living circle-five-minute living circle-residential neighborhood" [2]. Therefore, as the concept of "community life circle" has received attention, innovative thinking has been given to the functional connotation and construction mechanism of urban space. As a result, it has become an inevitable trend to improve community construction with the concept of "life circle".

The concept of "life circle" originated in Japan. In 1962, based on the central place theory, the theory of "life circle composition" was proposed, that is, life centers of different time scales were defined by setting circles with different radii and different population sizes according to a certain proportion, and it was used in the planning of Japan's territories. Subsequently, the concepts of "wide area living circle", "local living circle" and "settlement circle" were put forward. Among them, the concept of "settlement circle" became the prototype of the concept of "15-minute community life circle", and later this concept was accepted and gradually spread to other parts of Asia and South Korea and other countries. In European and American countries, the "neighborhood center" theory, the new urbanism paradigm and the urban theory related to Jane Jacobs theory all express the continuous transformation of community planning [3, 4]. From the "closed community" in the theory of "neighborhood unit" with car transportation as the main mode to the "open community" in the new urbanism paradigm with walking and public transportation as the main mode of transportation, this change also shows that the planning of the community life circle has gradually focused on openness and sharing, aiming to provide people with more complete life services [5]. In the post-epidemic era, despite the gradual loosening of measures to restrict regional activities to citizens, many cities still choose the concept of community life circle as the direction of urban recovery and reconstruction. For example, the Scottish government has taken measures to support the 20-minute community concept, that people can meet their needs within a 20-minute walk from home.

This concept enables people to live better and healthier lives and supports the goal of net zero [6]. In addition, the mayor of Paris also proposed the concept of a 15-minute city, that is, Parisians should be able to meet their shopping, work, entertainment and cultural needs within 15 minutes of walking or cycling. The newly released Mayors' Agenda for a Green and Just Recovery, released July 15 by C40 Cities mentioned that the 15-minute city concept may be the most concise and catchy way to repackage the idea as a pandemic economic recovery tool [7].

Research on the 15min-CLC in China mainly contains more theoretical discussions [8], such as the theory of 15min-CLC hierarchy construction, the theoretical guidance of 15min-CLC construction based on time geography or urban life circle theory [9], definition of different types of life circles [10] and qualitative research on resource allocation of public service facilities in 15 minutes based on urban land use [11]; in terms of empirical research, it contains some 15min-CLC evaluation research based on field survey and exploration data for specific types of basic service facilities (such as shopping or sports facilities) [12]. However, a city with a high quality of life needs a basic public service system with multiple types of coverage and public service guarantees covering all age groups. Therefore, the method and technical system for evaluating and optimizing the construction of a 15min-CLC in the city still need to be further explored [13]. In addition, the construction of the life circle should be closer to the daily needs of citizens, and the strength of community residents should be fully utilized to promote bottom-up community governance, and different infrastructure service facilities systems should be configured according to different needs [14]. However, in the existing studies focusing on the facility composition of the 15min-CLC, the basic service facilities in the 15min-CLC are often regarded as a uniform configuration, which ignores the different levels of community residents' needs for different types of basic service facilities. This phenomenon is mainly due to missing or incomplete data. In recent years, with the rapid development of Internet technology, especially the emergence of big data, some studies have begun to improve the comprehensiveness of the research framework and content of the community life circle, and began to build a multi-angle evaluation system and multi-scale empirical research [15], such as the study of the balance between the demand generated by residential areas and urban service provision under different spatial scales [16]. Shanghai is one of the earliest cities in China that put forward the concept of "15-minute community life circle". The "Shanghai 15-Minute Community Life Circle Planning Guidelines (Trial)" released in 2016 first proposed the concept of 15min-CLC, that is, it is equipped with basic service functions and public activity space required for life within a 15-minute walk, and forms a safe, friendly and comfortable social basic life platform [17]. Moreover, the "Shanghai Master Plan (2017–2035)" released in 2018 clearly proposes to build a 15min-CLC to provide appropriate housing for residents, create a more pleasant living environment, promote travel convenience for more people and give residents a higher sense of belonging and identity [18]. Shanghai, as the core city of the Yangtze River Delta urban agglomeration, the world-class urban agglomeration, often overlooks the creation of "daily life space" in the construction of its infrastructure service facilities due to excessive emphasis on large-scale projects [19], resulting in the unbalanced distribution of public service spaces and the neglect of the interests of disadvantaged groups [20]. Based on these issues, this study chooses Shanghai as a typical city, and combines the supply and demand of basic service facilities in the 15min-CLC to study the current situation of the 15min-CLC.

Based on the existing domestic and foreign theoretical and empirical research on the 15min-CLC, it can be seen that the planning of the "15-minute community living circle" not only needs to pay attention to the spatial distribution of the service scope of basic service facilities, but also the population in the 15min-CLC, not only the size of the population, but also the

characteristics of the population and the population's demand for basic service facilities. Therefore, this study introduces multi-source big data such as POI data, OSM data and Land-Scan population data to provide more accurate optimization suggestions for the construction of urban life circles. In summary, taking into account the supply and demand differences in the configuration of infrastructure service facilities in multiple types of 15min-CLC, this paper constructs a 15min-CLC multi-type basic service facility population matching measurement model and a 15min-CLC evaluation index system. Taking Shanghai as a typical city, through analyzing the current situation of the 15min-CLC in Shanghai, corresponding optimization suggestions are put forward. The method in this article also provides reference for future 15min-CLC research.

The rest of this article is as follows: Section 2 introduces the research area, research data (including POI data, OSM data and LandScan data set) and research methods. Section 3 introduces the research results in detail, including the model evaluation, the analysis of the current situation of the 15min-CLC in Shanghai and optimization suggestions. Section 4 includes the discussion which mainly explain the direction of further research. Finally, Section 5 is the conclusion of this research.

## Materials and methods

### Study area

The research area of this article is Shanghai, China ($30°40'$ to $31°53'$N, $120°52'$ to $122°12'$E) (Fig 1). The city has a total area of 6340.5 square kilometers, with 16 municipal districts including Huangpu, Xuhui, Changning, Jing'an, Putuo, Hongkou, Yangpu, Minhang, Baoshan, Jiading, Pudong New Area, Jinshan, Songjiang, Qingpu, Fengxian, and Chongming. The 7 districts of Yangpu, Huangpu, Xuhui, Changning, Jing'an and Putuo are collectively called the downtown area. There has been a rapid expansion in population size [21] the resident population has increased from 23.03 million in 2010 to24.24 million in 2018, nearly doubling in the past 30 years. In 2010, Shanghai's GDP was 249.22 billion dollars, and its per capita GDP based on the permanent population was 10826.78 dollars (based on current exchange rate). With Shanghai's sustainable high-quality development, by 2018, Shanghai's GDP was 494.4 billion dollars, and the per capita GDP calculated by the permanent population was 20423.6 dollars, nearly doubling in the past decade. As a result of the rapid population growth, Shanghai is facing a greater degree of mismatch between basic service facilities and population. Therefore, this article constructs a 15min-CLC infrastructure service facility population matching evaluation model, and completes a case study using Shanghai as a typical city, to improve the existing 15min-CLC research framework, and provide reference opinions for the sustainable development of cities and the promotion of smart cities.

### Data

**POI.** POI data is a kind of point data representing real geographic entities, and contains basic information such as name, address, latitude and longitude, and category. POI data has become more and more spatial information data source due to its easy access and rich information [22]. In China, AutoNavi Map not only has a large number of active users, but also has a more comprehensive coverage of basic service facilities. This article uses AutoNavi's 2018 POI data. As of the end of 2018, the total number of POIs in Shanghai was about 600,000. AutoNavi divides POI data into 20 categories based on its business services. According to the basic living needs of citizens, this article selects six categories of catering services, shopping services, life services, medical services, leisure services, and financial services. After data

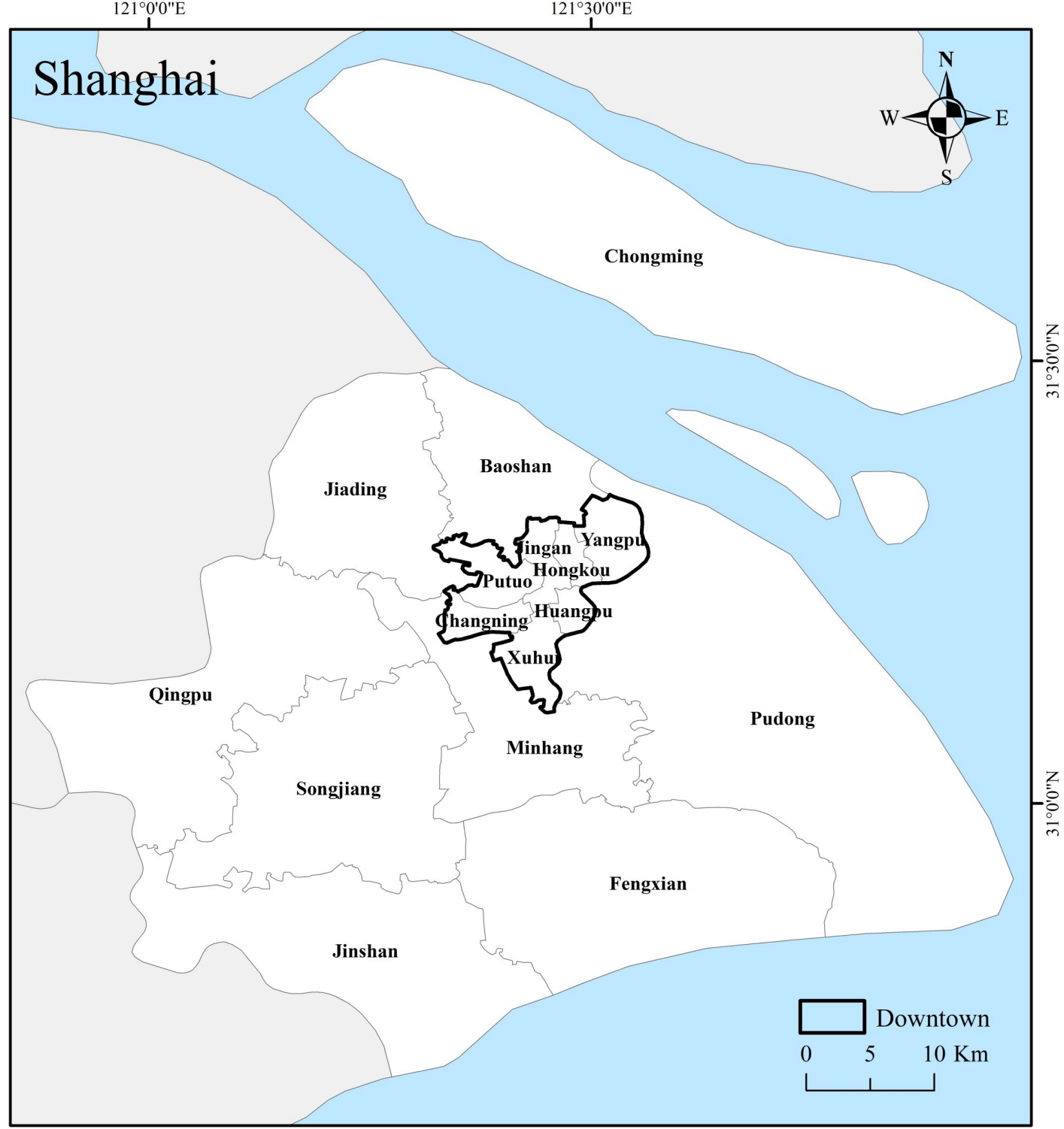

**Fig 1. Study area.** An overview of Shanghai. (Image Source: https://www.naturalearthdata.com/downloads/10m-cultural-vectors/).

preprocessing, about 200,000 data sets of basic service facilities based on the daily needs of citizens are finally obtained. The names and main contents of the categories correspond to Table 1.

**Table 1. Correspondence of basic service facility category names and main contents.**

| Category | Main contents | Count |
|---|---|---|
| Catering | Local specialty restaurants, cafes, tea restaurants, baking shops, etc. | 85837 |
| Shopping | Convenient shops, flower and bird markets, shopping malls, vegetable markets, etc. | 8681 |
| Life | Beauty salons, water service shops, laundry shops, etc. | 45826 |
| Medical | Emergency centers, community hospitals, general hospitals, specialist hospitals, pharmacies, etc. | 19721 |
| Leisure | Cinemas, parks, sports halls, KTV, senior activity rooms, etc. | 29041 |
| Financial | Bank, ATM, etc. | 9810 |

**Urban road network.**   OSM contains relatively complete urban traffic information, including highways, bicycle lanes, and trails, especially in cities with better development [23]. As a well-known volunteer geographic information, OSM data can be used free of charge through the OSM Open Database License (ODbL), which allows free and reasonable use of data for various applications. Its feature of real-time updates independently by users is also one of the reasons why it has gradually become the mainstream data source for urban traffic research. Many studies have proved the reliability of OSM data in different directions of urban traffic research, such as the study of pedestrian accessibility, the study of multi-modal path planning for the disabled, and the study of urban emergency evacuation. Therefore, this paper selects OSM road network data as the data source of urban road network. Based on the concept of a 15min-CLC, the pedestrian road network data in the OSM data set accessed at 2019 is selected as the urban road network data in this article. Through topology inspection and tailoring, we obtained the pedestrian road network in Shanghai in 2018.

**Population.**   This article uses the LandScan population raster dataset to express the spatial distribution of population. LandScan [24], developed by the US Department of Energy's Oak Ridge National Laboratory (ORNL), is a global population distribution dataset. LandScan combines local census data, road proximity, slope, land cover and other geospatial data and remote sensing images to determine population distribution through a specific mathematical model that meets specific regional conditions, with a spatial resolution of 1 km [25]. LandScan is currently free for educational groups and is updated annually. Compared with census data, LandScan has a shorter update cycle, and the display of population distribution is closer to reality, including areas that are not included in census data, such as work areas, leisure and entertainment areas and other areas without residential functions [26]. In this study, the 2018 LandScan global population data set was preprocessed to obtain the Shanghai 2018 population data distribution map (Fig 2).

## Methods

This article uses multiple types of POI data and road network data as basic data, combined with population data, and uses service area analysis combined with weight, to calculate the comprehensive service convenience of the 15min-CLC; And build a 15min-CLC measurement model and measurement index system, including the diversity of basic service facilities and population matching index of the 15min-CLC. This article evaluates the 15min-CLC at different spatial levels, and explores the current development and future development direction of the 15min-CLC.

**Weighted network service area analysis.**   The network service area analysis method is based on the road network and combined with certain impedance conditions (such as traffic mode, time or distance) to calculate the reachable coverage area, that is, the network service

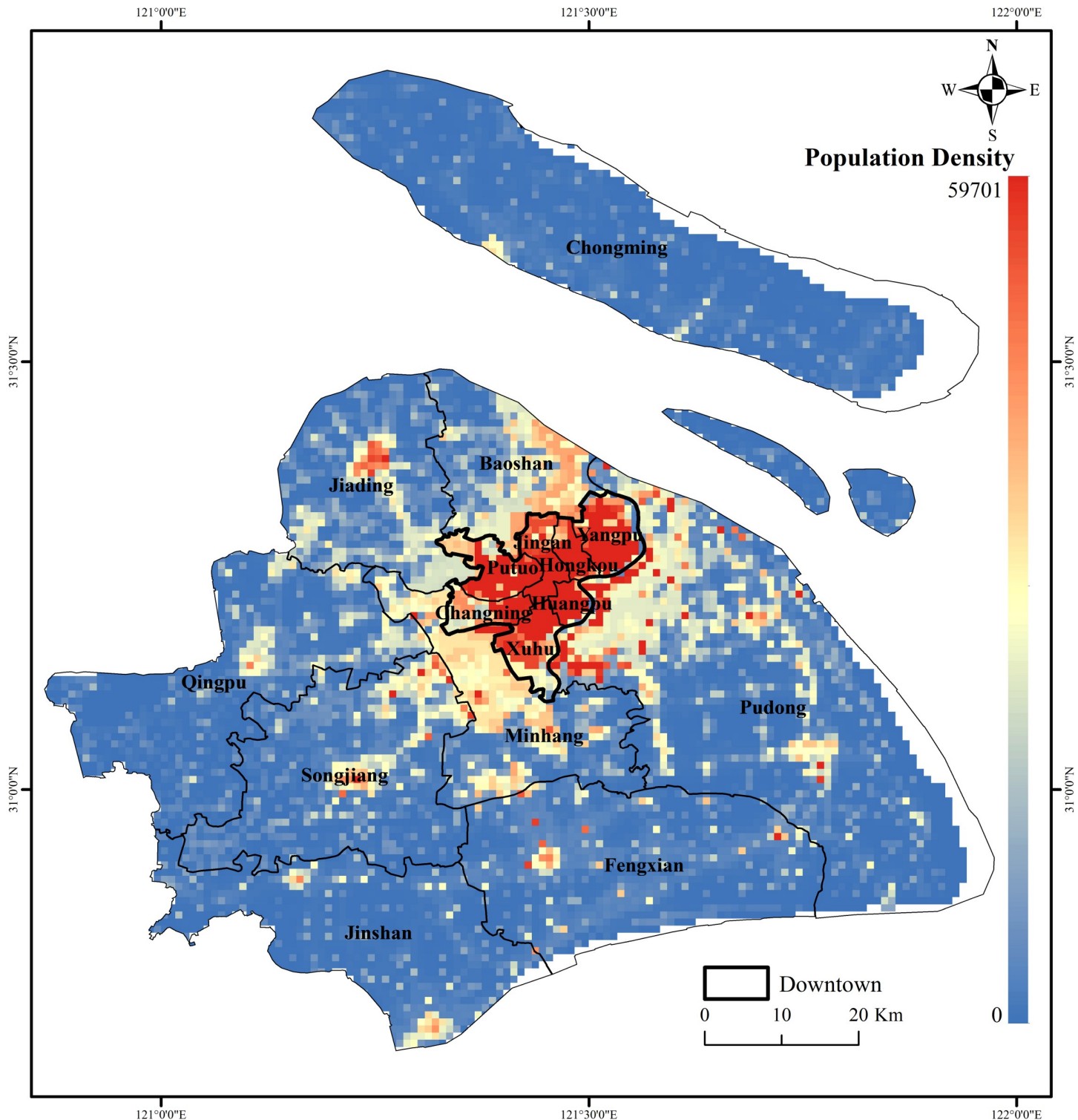

**Fig 2. Shanghai population distribution.** (Image Source: https://www.naturalearthdata.com/downloads/10m-cultural-vectors/).

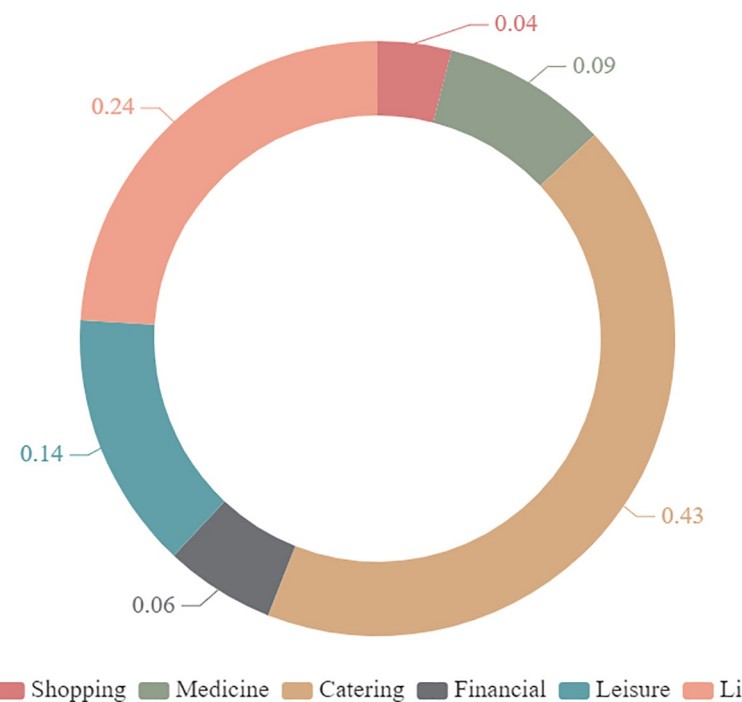

**Fig 3. Weight pie chart.** A description of the different levels of needs of different basic service facilities for citizens' daily life.

area [27]. Different from ordinary buffer analysis method [28], network service area analysis method receives network restrictions [29], so the result will be closer to reality.

Different types of POI data have different impacts on citizens' lives. In order to make the results more relevant to the daily needs of residents, when calculating the mixed service area, different weights are assigned to different categories of POI. This article uses a questionnaire to formulate a weighting table based on the different types of POI's impact on citizens' lives [30]. By publishing 500 online questionnaires (mainly for young and middle-aged groups between 20 and 65 years old) and 200 offline questionnaires (mainly for old people over 65 years of age), the applicants can rank different POIs according to daily needs. Among them, the most influential infrastructure service facility category was scored 6 points, and so on, a total of 672 valid questionnaires were received, including 487 online questionnaires and 185 offline questionnaires. Through the conversion of scores, the weight value of the impact of different types of basic service facilities on citizens' daily life is obtained (Fig 3). It can be seen that catering service facilities and life service facilities play an important role in ensuring basic services for residents.

The network limitation of the weighted network service area analysis method in this article is the urban pedestrian road network, and the impedance condition is 15 minutes walking distance (about 1 km), and then combined with the weights, calculate the reach of multiple types of infrastructure service facilities to complete the weighted network service area analysis. This result reflects the mixed accessibility of multiple types of infrastructure service facilities in the 15min-CLC. The higher the value, the better the comprehensive service convenience of multiple types of infrastructure service facilities in the 15min-CLC in the area.

**15min-CLC infrastructure service facilities compatibility measure model.** This paper constructs a 15min-CLC population supporting measure model and a measurement index system, which means calculating the mixed diversity of the basic service facilities of the 15min-

CLC through weighted kernel density analysis, and on the basis of this result, using the population supporting model to calculate the population matching index of the basic service facilities of the 15min-CLC.

*(1) Weighted kernel density estimation and 15min-CLC infrastructure service facilities mixed diversity index.* Kernel density estimation (KDE) calculates the density of point or line features based on specific search radius and quadrature kernel function through a moving cell, which is not affected by grid size and position [31]. The calculation results of this method can clearly express the distribution characteristics of single type basic service facilities in the 15min-CLC, which is calculated as follows:

$$D = \frac{1}{r^2} \sum_{i=1}^{n} [\frac{3}{\pi} \times \left(1 - \left(\frac{d_i}{r}\right)^2\right)^2]$$

$r$ represents the search radius, which is 1000 meters in this article; $i$ (the value range is the number of points within the search radius) represents any point within the search radius, and $d_i$ is the distance between the center point and point $i$.

A city with high quality of life must be a city that can meet the needs of urban citizens in all aspects, so the mixed diversity of infrastructure service facilities in the 15min-CLC has become a crucial evaluation indicator [32]. With reference to the diversity index in biology, similar to the analysis of the weighted network service area, this paper uses the linear weighting method to assign weights to the calculation results of the kernel density of different types of infrastructure service facilities, which can obtain the mixed diversity index (DM) of the infrastructure service facilities of the 15min-CLC in a certain area (refers to the 100-meter grid in this article) and the calculation formula is as follows:

$$\text{DM}_i = \sum_{k=1}^{6} D_{ik} * W_k$$

$D_k$ is the density of the k-th POI in the 15min-CLC calculated by kernel density estimation; $W_k$ represents the weight of the k-th POI. This result can reflect the diversity of basic service facilities in a 15min-CLC. The larger the value, the richer the types of basic service facilities in the area.

*(2) Population matching measurement model and 15min-CLC infrastructure service facility population supporting index.* A city with a comfortable living environment not only requires a rich variety of high-density urban infrastructure service facilities, but also requires the configuration of basic service facilities to match the population. If high-density basic service facilities are deployed in areas with low population, the citizens will feel the crowdedness of the city and reduce the comfort of life. Therefore, this paper designs a 15min-CLC infrastructure service facility population matching measurement model, including calculating the population matching degree of a single type of infrastructure service facilities in a certain area, and obtaining the population matching index of multiple types of infrastructure service facilities in a certain area through linear weighted summation, the calculation formula is as follows:

$$I_i = \frac{DM_i}{P_i}$$

Among them, $P_i$ represents the population density in a certain area.

This index can better reflect the population matching of basic service facilities in the 15min-CLC. The larger the value, the better the compatibility. Through this model, it can

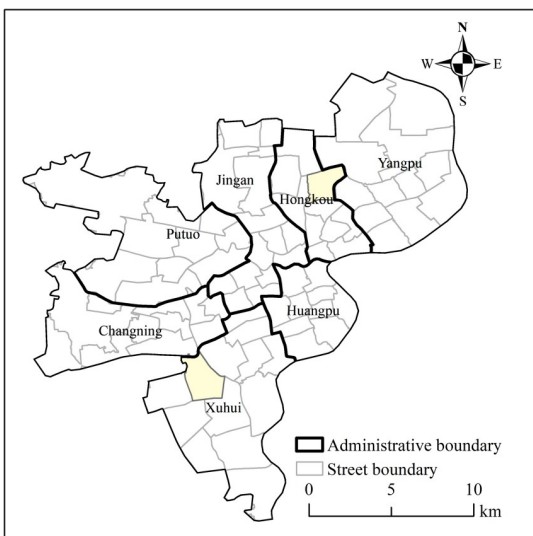
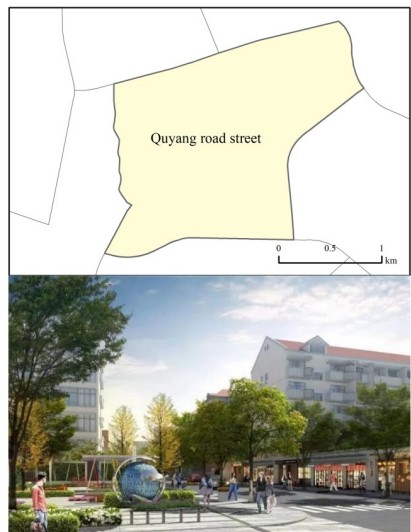
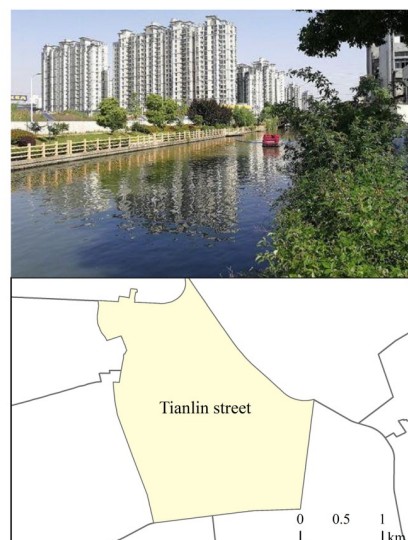

**Fig 4. Description of the two cases.** Tianlin Street and Quyang Road Street, including the location of the two cases and pictures of the scene. (Image Source: https://www.naturalearthdata.com/downloads/10m-cultural-vectors/).

better provide data support and reliable suggestions for the future development trend of 15min-CLC.

**15min-CLC compatibility measure model evaluation.** In order to validate the accuracy and applicability of the model, the study uses data processing method for model verification which means to compare the consistency between the actual test results and the model output results under the same test conditions to evaluate the credibility of the model. Considering the operability of model verification, this article selects familiar streets and combines actual conditions to compare street statistics with model calculation results to complete the model verification. Based on the positioning area of daily life and learning, this paper chose Tianlin Street and Quyang Road Street (Fig 4). At the same time, these two streets are also the first batch of pilot streets in the 15min-CLC in Shanghai.

Quyang Road Street is located in the northeast of Hongkou District, Shanghai, with a population of 102,564 (based on census data), an area of 3.05 square kilometers, and a total of 6,881 infrastructure service facilities in the category of concern in this article. Its planning vision is guided by "innovation, coordination, green, open, and sharing", by creating a higher-quality community environment, a more dynamic creative space, and higher-quality supporting services, it will promote the upgrading and become a model of quality community for the renewal and development of Shanghai City. Tianlin Street is located in the west of Xuhui District, Shanghai, with a population of 97,171, an area of 4.19 square kilometers, and a total of 7,249 infrastructure service facilities. Its planning vision is to connect the Puhuitang Riverside Slow Traffic System, promote the integration between the community and the urban green valley, create a shared happy community.

## Analysis of the status quo of 15min-CLC in Shanghai

### Analysis of 15min-CLC measurement model evaluation

The result of Tianlin Street based on street statistics is 0.0281, and the result of model calculation is 0.0284, with an accuracy of 98.94%; the result of Quyang Road Street based on street statistics is 0.0215, and the model calculation result is 0.0221, with an accuracy of 97.3% (Table 2).

**Table 2. Model validation result.**

| Street name / Attributes | Tianlin Street | Quyang Road Street |
|---|---|---|
| Population(person) | 97171 | 102564 |
| Facility | 7249 | 6881 |
| Area(km$^2$) | 4.19 | 3.05 |
| Index(actual) | 0.0281 | 0.0215 |
| Index(calculated) | 0.0284 | 0.0221 |

The average accuracy is 98.1%, indicating that the model has high accuracy and is suitable and reasonable for the research content of this article. Through model verification, it also shows that the model conforms to the planning vision of the 15min-CLC in Shanghai: to build a high-quality community, that is, to create a high-quality community environment with facilities and people-oriented, that is, a more dynamic innovation space, a higher-quality basic service system, which means that citizens can enjoy relatively complete medical, commercial, cultural and sports facilities and other basic public service facilities within a 15-minute walk from home.

## Analysis of comprehensive service convenience of multiple types of infrastructure service facilities in the 15min-CLC

This paper first regards the 6 types of infrastructure service facilities as a balanced configuration, performs network service area calculations and directly superimposes the sum, and obtains the reachable range distribution of multiple types of infrastructure service facilities in 15min-CLC without weighting. From the perspective of the service coverage rate of different types of infrastructure service facilities (Table 3), the service coverage rate of catering service facilities and leisure service facilities is relatively high; the service coverage rate of medical service facilities reaches 76.07%; the service coverage rates of life services and shopping service facilities were 67.47% and 62.36%, respectively, and the service coverage rates of financial service facilities were only 41.63%. However, citizens have different levels of demand for different infrastructure service facilities, and there is often no need for too high coverage for the types of infrastructure service facilities with low demand. Therefore, this paper assigns weights to the network service area results of single type of infrastructure service facilities and then superimposes them again to obtain the mixed reach distribution of multiple types of infrastructure service facilities in the 15min-CLC. The closer the value of a certain area is to 1, the better the comprehensive service convenience of infrastructure services in the area, that is, the more convenient it is to live in the area (Fig 5). The figure shows that the comprehensive service convenience of basic service facilities in the downtown area is higher, that is, more convenient to reach basic service facilities. This result shows that the rationality of the overall distribution of urban infrastructure service facilities in Shanghai still needs to be optimized, and that the accessibility of different types of infrastructure service facilities also has different optimization spaces. Considering the impact of different types of infrastructure service facilities on citizens' lives and their service coverage, Shanghai can pay more attention to optimizing the location of life service facilities and shopping service facilities.

**Table 3. Coverage rate of various basic service facilities.**

| | Catering | Shopping | Life | Medical | Leisure | Financial |
|---|---|---|---|---|---|---|
| Service Coverage (%) | 87.86 | 62.36 | 67.47 | 76.07 | 89.36 | 41.63 |

### Analysis on matching index of basic service facilities in the 15min-CLC in Shanghai

Based on the POI data and combined with the weight value, this paper uses the weighted kernel density estimation to calculate the 15min-CLC multi-type infrastructure service facility mixed diversity index. The diversity of infrastructure service facilities in the downtown area is high, and the overall trend is to decay radially outward from the center. At the same time, there are areas with high diversity index in local locations (such as Disney, Songjiang University Town, etc.). However, citizens in different regions have different demands for different infrastructure service facilities. For example, many living service facilities are not needed in the scenic area, and perhaps the number of catering services in the downtown area is far from enough to meet the life needs of the downtown people. Moreover, with the continuous improvement of people's pursuit of quality of life, the configuration of urban infrastructure service facilities that do not match the number of populations cannot bring a suitable life to the citizens, so the calculation of the diversity of infrastructure service facilities and the population supporting index becomes important. Therefore, this paper calculates 15min-CLC infrastructure service facilities population supporting index and the result is shown in Fig 6. The more middle the value is, the more the basic service facilities of the place are matched with the population of the place, that is, the more reasonable the configuration. The population distribution of different regions and the population supporting index of basic service facilities in the region are shown in Table 4. The value of this index should not be too large, which will result in a phenomenon of over-density of basic service facilities; if the value of this index is too small, it will not be able to meet the daily needs of citizens. It can be seen from the figure that the index of the area enclosed by the yellow ellipse that contains most of the downtown area is mostly in the middle section, while the index of the area enclosed by the blue ellipse has more high value areas. This result shows that the basic service facility configuration of the 15min-CLC in the downtown area is more compatible with the population distribution. In the peripheral areas of the urban area (mainly including Baoshan District, Jiading District, Minhang District, Songjiang District, and Pudong New Area), there is a phenomenon of excessive construction of infrastructure service facilities. In other areas (mainly including Qingpu District, Jinshan District and Fengxian District), due to the uneven population distribution, the spatial layout of infrastructure service facilities still needs to be further optimized. With the development of the five major new cities in Shanghai and the development of new rail transit lines in Shanghai, the spatial layout of infrastructure services outside the yellow circle area can also be optimized in combination with this background to better meet the current development model of Shanghai and the current population distribution characteristics.

## Optimization policy of the 15min-CLC

### 15min-CLC optimization strategy based on the perspective of supply improvement

For the optimization of basic service facilities in different regions and categories, supplementary construction of basic service facilities should be carried out in uncovered areas or areas with low comprehensive service coverage, so as to improve the supply of basic service facilities in the 15min-CLC. The construction of basic service facilities in the urban fringe and urban-rural areas should be focused on, so as to improve the full coverage of the basic service facilities. While ensuring the reasonable and economical service scope of the basic service facilities in the 15min-CLC in the downtown area, the number of basic service facilities in the downtown area can be appropriately reduced, while the quantity or quality of the basic service

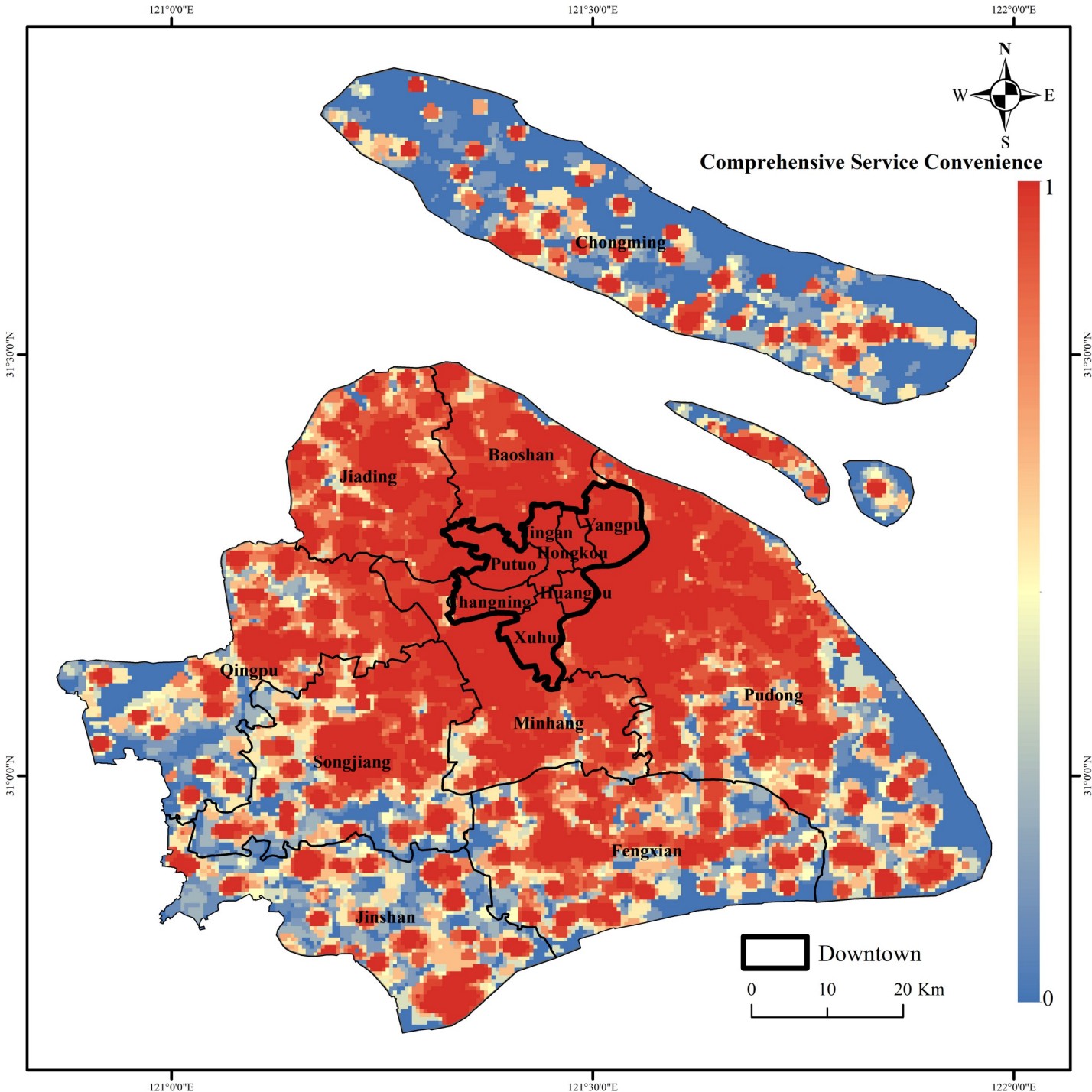

**Fig 5. Distribution of mixed accessibility of basic service facilities in the 15min-CLC in Shanghai.** (Image Source: https://www.naturalearthdata.com/downloads/10m-cultural-vectors/).

facilities in the urban fringe and the urban-rural junction can be improved. For areas that are not reachable by foot, the pedestrian road network can be reconstructed to appropriately break the existing community boundaries, and the supply of basic service facilities within a 15-minute walk has been guaranteed.

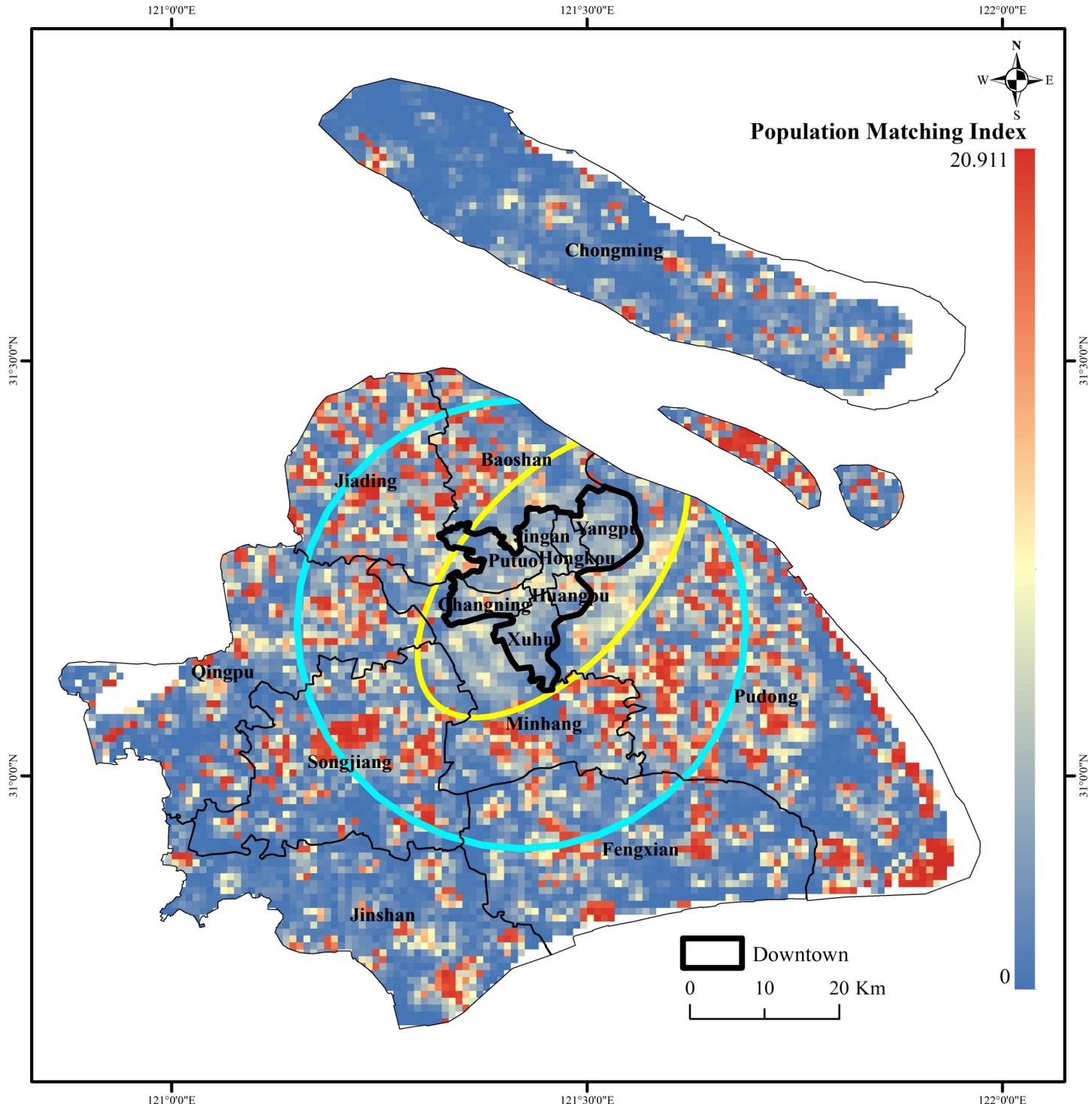

**Fig 6. Distribution of facility population supporting index in the 15min-CLC in Shanghai.** (Image Source: https://www.naturalearthdata.com/downloads/10m-cultural-vectors/).

From the perspective of different types of basic service facilities, the shortcomings of life service facilities and shopping facilities are more obvious. With the e-commerce era and the popularization of shared infrastructure, traditional shopping facilities and life service facilities

**Table 4. Population matching index and population.**

|  | Downtown | Fengxian | Chongming | Pudong | Jinshan |
|---|---|---|---|---|---|
| **Population Matching index** | 0.9934 | 6.0350 | 7.8034 | 1.2922 | 8.7326 |
| **Population (ten thousand person)** | 632.9574 | 108.3463 | 70.3722 | 504.443 | 73.241 |
|  | **Qingpu** | **Minhang** | **Jiading** | **Songjiang** | **Baoshan** |
| **Population Matching index** | 6.1106 | 2.6981 | 4.6032 | 4.2305 | 3.4653 |
| **Population (ten thousand person)** | 108.1022 | 242.9372 | 147.1231 | 158.2398 | 190.4886 |

have been impacted, and people's lifestyles and consumption patterns have changed, such as contactless delivery in the epidemic. As a result, the supply scale of shopping service facilities and life service facilities in the community life circle can be improved by building a community business cycle, creating a comprehensive service-oriented community that covers the entire community, and promoting the development of smart community commerce [33]. The implementation of the community business model has also been initially explored at home and abroad. In China, it is mainly manifested in the community group buying business model, that is, the community group buying model in which major e-commerce platforms are equipped with community leaders, such as Suning Tesco's community group buying system consisting of platform warehouses, group leader sites and residents' homes. The huge retail market and distribution chain also provide a good foundation for the community business model. In foreign countries, related urban design concepts have also been proposed. For example, in the Netherlands, 'Hyperlocal Micromarket' was proposed, which is a simple 16-square grid design for a tiny marketplace that can be quickly and cheaply assembled in public squares, allowing people to shop local while also following social distancing guidelines [34]. The community business model can not only increase the frequency of use of physical shopping service facilities in high-saturation areas, but also ease the situation of insufficient resources in second- and third-tier cities. In the future, the hierarchical spatial layout of physical shopping service facilities can be optimized according to the status quo of different regions and different scales to improve the quality, quantity and use efficiency of shopping service facilities.

For medical service facilities, although its service coverage rate reaches 76.07%, in Shanghai where the aging problem is increasingly prominent, the scale of medical service facilities in the 15min-CLC, especially medical service facilities and elderly care facilities, still needs to be improved. In the case of uneven distribution of medical service facilities, improve the quality of medical service facilities in the downtown area. For other areas, especially the urban fringe areas such as Qingpu District, Fengxian District, and Jinshan District, focus on expanding medical service facilities and increasing the service scale of community medical service facilities. At the same time, for the construction of elderly care facilities, it can be configured together with medical service facilities through the collocation model of medical service facilities and elderly service facilities.

In summary, the suggestions from the perspective of the supply of the 15min-CLC are mainly to make the supply of basic service facilities in the 15min-CLC more balanced through functional replacement or urban renewal design.

## 15min-CLC optimization strategy based on the perspective of demand matching

The downtown area is more consistent with population matching than other districts on the whole, but it has not yet reached the most suitable situation, and the possibility of land expansion in the downtown area is relatively small. In the future, it is possible to improve the supply

and demand matching of the 15min-CLC through the construction of functional integration areas. By integrating basic service functions for different age groups in the same basic service facility, that is, in a way of shared use and common layout, the efficiency of facility use and the quality of basic service facilities in the 15min-CLC are improved. For the downtown area, it is also possible to break the original community boundaries, integrate adjacent communities, and supplement the construction of a community walking road network to improve the demand matching of the 15min-CLC. For the population matching of infrastructure service facilities in other regions, not only the quality of the infrastructure service facilities should be paid attention to, but also the scale of citizen demand should be ensured as the population distribution changes due to the expansion of the five major new cities and Shanghai's road network. In the future, by optimizing the hierarchical infrastructure service facility system, the population supporting facilities in the 15min-CLC can be improved. Taking medical facilities as an example, the existing service levels of community medical facilities often fail to meet the daily needs of citizens, that is, improving the level of community medical services, thereby increasing the efficiency of community medical facilities, and solving the contradiction between supply and demand of medical service facilities.

The functional sharing and opening of community space have always been the direction of the 15min-CLC. The Scottish government has spent £39m on temporary infrastructure to enable people to walk and cycle while maintaining social distancing [35]. In the future, for various regions at home and abroad, the "spatial time-sharing multiplexing" model can be introduced, which means that the needs of basic service facilities corresponding to the daily habits of people of different age groups, including lifestyle, travel time period, and time period for using different types of basic service facilities, are integrated in different time periods in the spatial layout, that is, to plan a 15min-CLC in a way of shared use and joint planning.

## Discussion

The research area of this article mainly focuses on Shanghai as a whole, and has begun to pay attention to the relationship between human needs and the services provided by the basic service facilities in the 15min-CLC, but there are still many aspects to be further studied.

The case study area of this article focuses on the whole in Shanghai and its administrative divisions, and no detailed attention to the different communities or community plaza street, but different regions have different configuration requirements and policies (such as ecological protection policies on basic services in Chongming area). Even different communities in the city center have different requirements for basic service facilities. Moreover, in Shanghai, there are often new and old communities. For new communities, it is only necessary to optimize the construction of the 15min-CLC through some community micro-update projects, such as the Lujiazui community in Pudong New Area [36]; for some old communities, it is necessary to form a renovation plan through overall evaluation, such as Putuo Wanli Community.

From the perspective of data, the urban pedestrian road data source used in this article, OSM data, is already a relatively mature volunteer geographic information service, but its data still has a certain degree of subjectivity, and the topology and complete coverage of road data still need to be further screened; as for the population data used in this article, the LandScan data set has been able to better represent the distribution of the resident population, but its spatial resolution still needs further consideration for the promotion and application of 15min-CLC research in other cities; the POI data used in the article does not contains all types, but only filters some categories of basic service facilities related to the basic needs of life. However, with the continuous improvement of people's demand for quality of life, we should also

consider building an open and vigorous 15min-CLC, paying attention to the needs of people in the life circle for health, fitness, culture, and communication.

Although this article has begun to pay attention to groups of different populations, the relationship between the needs of different service facilities corresponding to different groups of the population and the construction of 15min-CLC still needs to be further explored. Groups of different age levels, and even groups of different characteristics (such as office workers, student groups), have different requirements for the configuration of basic service facilities in the 15min-CLC. For example, on working days, office workers pay more attention to the basic service facilities that can be opened at night, such as a 24-hour convenience store; while older people are different, they need basic service facilities based on family services and leisure and entertainment, such as supermarkets, vegetable markets, and activity rooms. In future research, age should be used as the main factor to divide the needs of different groups of people for different basic service facilities. In Shanghai, the allocation of public service facilities should further focus on the needs of vulnerable groups such as the elderly and children. In communities dominated by the elderly (for example, the elderly population over 60 years old reaches more than 25% of the total population of the community or the elderly population over 80 years old accounts for more than 10%), the use of basic service facilities in the 15min-CLC can be adjusted at different times according to the walking ability, needs and lifestyle of the elderly [37]. For communities with a high proportion of children (which can be determined by birth rate), children's participation in the supporting infrastructure services should be considered, such as the safety and future development of the infrastructure services for children [38].

In 2020, we encountered COVID-19. After the epidemic, the construction of the 15min-CLC also had an impact. For example, during the epidemic, citizens purchase more daily necessities through contactless delivery. After the epidemic, e-commerce factors (such as food delivery and express delivery) should also be considered as one of the factors to be considered for optimizing the construction of a 15min-CLC. In addition, through the epidemic, we once again realize the importance of smart communities, especially healthy smart communities. Many studies have proved that the construction of healthy smart communities will affect the citizens' motivation for sports in the community, and then affect the citizens' health [39]. The WELL Community Standard is also the first to be released as a global standard for trial operation, proposing community standards based on health, inclusiveness, fairness, comprehensiveness and vitality [40]. In West England, there have also been studies on built-up factors and natural environment in communities and human health [41]; Domestically, China's health community standards have gradually formed (as shown in Fig 7). Therefore, the construction of basic service facilities for the 15min-CLC should not only consider the basic living needs of community residents. With urban development and people's pursuit of higher quality of life,

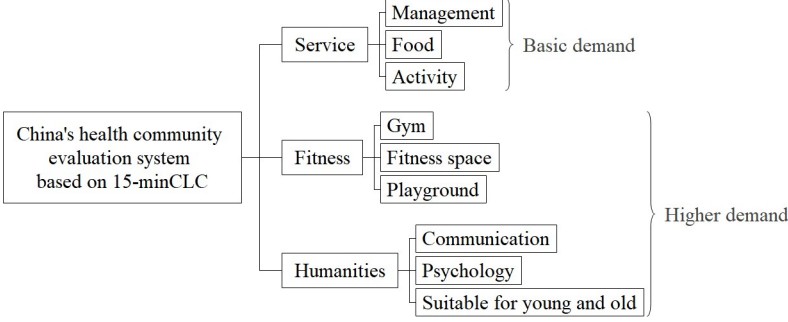

**Fig 7. China's health community evaluation system based on 15min-CLC.**

spiritual pursuits should also be included in the consideration of supporting infrastructure service facilities in the 15min-CLC, such as fitness-related facilities (such as community fitness spaces) and cultural exchange facilities (such as community exhibition halls, community theaters, etc.).

This article proposes a 15min-CLC infrastructure service facility population matching measurement model based on big data sources (including POI data, urban road network data, and active population distribution data). Based on this model, a smart community system that improves residents' experience can be built, which can include smart monitoring, smart distribution, smart medical treatment, etc., so as to promote the smart management of various public service facilities in the community and enable the community to have a higher effect on emergencies.

## Conclusion

This article uses urban multi-source spatial data sets, combined with spatial analysis models and statistical models such as network analysis, entropy, linear weighting, and kernel density analysis to construct a 15min-CLC evaluation model, and summarize existing shortcomings based on the evaluation results and propose corresponding optimization strategies. The existing shortcomings are mainly manifested in the following aspects: (1) From the perspective of classified basic service facilities, the spatial construction of Shanghai's overall life service facilities and shopping service facilities needs to be optimized. (2) From the perspective of comprehensive evaluation, the comprehensive service convenience of basic service facilities in the downtown area is relatively high, while the comprehensive service convenience of urban infrastructure service facilities in the suburbs and outer suburbs is relatively low. The diversity of basic service facilities in the 15min-CLC in the downtown area is relatively consistent with the population distribution, while the peripheral areas of the urban area (mainly including Baoshan District, Jiading District, Minhang District, Songjiang District and Pudong New Area) have an excess of infrastructure service facilities.; And other areas (mainly including Qingpu District, Jinshan District and Fengxian District) due to uneven population distribution, the spatial layout of their basic service facilities still needs to be further optimized. Based on the above shortcomings, the following optimization strategies are proposed from the perspective of supply optimization: (1) For life service facilities and shopping service facilities, the development of community business models can be used to promote the development of new life service facilities and shopping service facilities. (2) For the optimization of basic service facilities in different regions or different types, the number of basic service facilities in the downtown area can be appropriately reduced, and the construction of basic service facilities in the urban fringe and urban-rural areas should be focused on, and the full coverage of the basic service facilities should be improved. (3) In order to face the aging characteristics of Shanghai, for medical service facilities, community medical institutions can be improved by means of facility function conversion, merger and reconstruction, etc. From the perspective of demand matching, the following optimization strategies are proposed: (1) For the downtown area, construct or break the original community boundary through functional integration areas, integrate adjacent communities, and supplement the construction of a community walking road network to improve the demand matching of the 15min-CLC. (2) For other districts, optimizing the hierarchical infrastructure service facility system can improve the population matching of infrastructure service facilities in the 15min-CLC.

This article uses multi-source big data (including POI data, urban road network data) to study the 15min-CLC, which can better avoid the problems of small sample size and low time and space resolution in the existing 15min-CLC research due to the use of statistical data and

questionnaires as data sources, making the research results more universal and representative. The 15min-CLC index system designed in this paper takes into account the diversity of service infrastructure and population matching in the 15min-CLC in the whole or at different levels, that is, the indicators of both supply and demand are considered at the same time so that the result can provide reference opinions for the next step of the 15min-CLC in Shanghai and this method can be extended to the whole country.

As the concept of the 15min-CLC continues to deepen its influence in the Chinese urban planning system, the construction framework of the 15min-CLC is constantly being explored and optimized. The optimization of the 15min-CLC not only needs to pay attention to existing problems and formulate corresponding optimization strategies, but also requires the cooperation of various fields; while improving the spatial layout of basic service facilities, it also needs to develop good management methods to ensure 15min-CLC planning and construction. The construction of 15min-CLC at different scales will be the focus of future research. Therefore, we should gradually deepen the integration of multidisciplinary knowledge, deepen the participation of residents as the main body of the community, and encourage residents to actively participate in the construction of the living environment, which will become more and more important in the future 15min-CLC planning. This paper builds a 15min-CLC evaluation model based on the "people-oriented" concept, and uses Shanghai as an example to carry out research, and proposes corresponding optimization strategies based on supply and demand matching and summarizing existing shortcomings. This article attempts to explore a replicable 15min-CLC planning model, so that it can be extended to the Yangtze River Delta urban agglomeration, to provide reference for further research on the 15min-CLC, and to promote urban construction under the concept of sustainable development.

## Author Contributions

**Writing – original draft:** Haoyuan Wu.

**Writing – review & editing:** Liangxu Wang, Zhonghao Zhang, Jun Gao.

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
