## [Decision Letter · Decision Letter 0]

23 Jun 2021

PONE-D-21-14500

Analysis and optimization of 15-minute community life circle based on supply and demand matching: A case study of Shanghai

PLOS ONE

Dear Dr. Wang,

Thank you for submitting your manuscript to PLOS ONE. After careful consideration, we feel that it has merit but does not fully meet PLOS ONE’s publication criteria as it currently stands. Therefore, we invite you to submit a revised version of the manuscript that addresses the points raised during the review process.

We look forward to receiving your revised manuscript.

Kind regards,

Feng Chen

Academic Editor

PLOS ONE

2. We note that Figures 1,2,4,5 and 6 in your submission contain [map/satellite] images which may be copyrighted. All PLOS content is published under the Creative Commons Attribution License (CC BY 4.0), which means that the manuscript, images, and Supporting Information files will be freely available online, and any third party is permitted to access, download, copy, distribute, and use these materials in any way, even commercially, with proper attribution. For these reasons, we cannot publish previously copyrighted maps or satellite images created using proprietary data, such as Google software (Google Maps, Street View, and Earth). For more information, see our copyright guidelines: http://journals.plos.org/plosone/s/licenses-and-copyright.

   a. You may seek permission from the original copyright holder of Figures 1,2,4,5 and 6 to publish the content specifically under the CC BY 4.0 license. 

Reviewers' comments:

Reviewer's Responses to Questions

**Comments to the Author**

1. Is the manuscript technically sound, and do the data support the conclusions?

Reviewer #1: Yes

Reviewer #2: Yes

2. Has the statistical analysis been performed appropriately and rigorously? 

Reviewer #1: Yes

Reviewer #2: Yes

3. Have the authors made all data underlying the findings in their manuscript fully available?

Reviewer #1: Yes

Reviewer #2: Yes

4. Is the manuscript presented in an intelligible fashion and written in standard English?

Reviewer #1: Yes

Reviewer #2: Yes

5. Review Comments to the Author

Reviewer #1: It is an interesting paper analyzing the 15-minute community life cycle in Shanghai. The research is well-conducted; however, some key concerns need to be solved before its publication.

1.The research background is kind of insufficient. The readers may need to know more about the concept of the “15-minute community life cycle”. Maybe adding a paragraph describing current policies and similar policies in other cities/countries may help the readers better understand the concept. I would also recommend the authors to add more content in key factors of the concept (for example, what kinds of facilities are important for this concept? It might be better to explain the design of the models), so the readers will have no problems in understanding the following analysis.

2.It may be clearer to add a data section in materials and methods to introduce the data sources systematically.

3.This research has the potential to be further developed, while the experience in Shanghai may be suitable for other cities in China or even in other countries. Maybe the author could add more general discussions on policy implications that are not limited to Shanghai's local context.

Reviewer #2: This paper evaluates Shanghai’s 15min-community life circle (15min-CLC) from the perspective of supply and demand matching. Based on multisource big dataset, they analyze comprehensive service convenience of multiple types of infrastructure service facilities and calculate their supply-demand matching index. The paper is meaningful, since it provides an empirical basis, which follows the buzzwords of people-oriented urban construction and sustainable urban development. However, some detailed need to be revised.

1. Global knowledge contribution: PLOS ONE is an international journal. This paper needs to engage with the wider readership of the journal. How to link the findings/conclusions in this paper with the previous findings/conclusions from other countries?

2. In the method section. It is unnecessary to explain the meanings of Wk and Dk twice.

3. The color distribution of Figure 5 is unclear. Please change the color to make it easy to identify areas with high, medium, and low values.

6. PLOS authors have the option to publish the peer review history of their article (what does this mean?). If published, this will include your full peer review and any attached files.

Reviewer #1: No

Reviewer #2: No

---

## [Author Response · Author response to Decision Letter 0]

3 Aug 2021

Responds to journal requirements:

As required by the journal, all pictures and content should be published in compliance with CC4.0. Therefore, we replaced the boundary data of the maps included in the article (Figures 1.2.4.5 and 6) with open-access Nature Earth vector boundary data and added a text description of the source of the image in the corresponding title of the image.

Responds to the reviewer’s comments:

Reviewer #1:

1. Response to comment: 

The research background is kind of insufficient. The readers may need to know more about the concept of the “15-minute community life cycle”. Maybe adding a paragraph describing current policies and similar policies in other cities/countries may help the readers better understand the concept. I would also recommend the authors to add more content in key factors of the concept (for example, what kinds of facilities are important for this concept? It might be better to explain the design of the models), so the readers will have no problems in understanding the following analysis.

Response: 

After considering your suggestions, we re-written the introduction, including its content and the order of paragraphs. In the first paragraph, we have supplemented the description of the concept of Japanese community life circle, such as ‘that is, life centers …… according to a certain proportion,’; We have also expanded the description of the development of theoretical research on community life circles in European and American countries in the second paragraph, such as ‘In European and American countries, …… with more complete life services[5].’, and other existing policies related to the 15-minute community living circle, such as ‘In the post-epidemic era, …… as a pandemic economic recovery tool[7].’ In addition, as you suggested, we have also added a description of the factors that should be emphasized in the concept of 15-minute community living circle, such as ‘Based on the existing domestic and foreign theoretical and empirical research on the 15-minute community living circle, ……, but also the characteristics of the population and the population's demand for basic service facilities.’ in the penultimate paragraph of this section.

2. Response to comment: It may be clearer to add a data section in materials and methods to introduce the data sources systematically.

Response: 

After considering your suggestions, we think it is very appropriate to write the description of the data in the method and material section of the article as a separate subsection. Therefore, in the materials and methods section, we merged the original three parts of "POI", "urban road network" and "population" into the "data" subsection. We also added detailed descriptions to the existing data descriptions in the text.

3. Response to comment: 

This research has the potential to be further developed, while the experience in Shanghai may be suitable for other cities in China or even in other countries. Maybe the author could add more general discussions on policy implications that are not limited to Shanghai's local context. 

Response: 

After considering your suggestions, we have supplemented the relevant content in the optimization policy section, and have also made corresponding changes to the relevant titles of the chapters. In the ‘15min-CLC optimization strategy based on the perspective of supply improvement’ part, we added ‘The implementation of the community business model has also been initially explored at home and abroad. ……, quantity and use efficiency of shopping service facilities.’ to illustrate the domestic and foreign discussions on the development of community business models and the relevant recommendations of this article on the promotion of this model; In the next part, we supplemented the existing case of the 15-minute community life circle open sharing plan, and expounded on the implementation of the suggestions proposed in this article for the optimization of the existing plan, such as ‘The functional sharing and opening of community space have always been the direction of the 15min-CLC. ……, to plan a 15-minute community living circle in a way of shared use and joint planning.’.

Reviewer #2:

1. Response to comment: 

This paper needs to engage with the wider readership of the journal. How to link the findings/conclusions in this paper with the previous findings/conclusions from other countries?

Response: 

As you suggested, we have added some existing policies and cases of other countries regarding the 15-minute community living circle in the optimization policy section, and supplemented the future optimization directions for these policies and cases proposed in this article, such as ‘In the future, the hierarchical spatial layout of physical shopping service facilities can be optimized ……, quantity and use efficiency of shopping service facilities.’ in the part of ‘15min-CLC optimization strategy based on the perspective of supply improvement’ and ‘In the future, for various regions at home and abroad, ……, to plan a 15-minute community living circle in a way of shared use and joint planning.’ in the part of ‘15min-CLC optimization strategy based on the perspective of demand matching’.

2. Response to comment: 

In the method section. It is unnecessary to explain the meanings of W_k and D_k twice.

Response: 

We are very sorry for the negligence of the repeated explanation in the text. For this we have kept the explanation of the parameters W_k and D_k that appeared at the beginning (in the ‘Weighted kernel density estimation and 15min-CLC infrastructure service facilities mixed diversity index’), and deleted the explanation of W_k and D_k that appeared in the "Population matching measurement model and 15min-CLC infrastructure service facility population supporting index".

3. Response to comment: 

The color distribution of Figure 5 is unclear. Please change the color to make it easy to identify areas with high, medium, and low values.

Response: 

It is really true as you suggested that the color distribution in Figure 5 should be optimized. Therefore, we changed the color bar, adjusted the color configuration, and remade Figure 5.

---

## [Decision Letter · Decision Letter 1]

18 Aug 2021

Analysis and optimization of 15-minute community life circle based on supply and demand matching: A case study of Shanghai

PONE-D-21-14500R1

Dear Dr. Wang,

We’re pleased to inform you that your manuscript has been judged scientifically suitable for publication and will be formally accepted for publication once it meets all outstanding technical requirements.

Kind regards,

Feng Chen

Academic Editor

PLOS ONE

Additional Editor Comments (optional):

Reviewers' comments:

Reviewer's Responses to Questions

**Comments to the Author**

1. If the authors have adequately addressed your comments raised in a previous round of review and you feel that this manuscript is now acceptable for publication, you may indicate that here to bypass the “Comments to the Author” section, enter your conflict of interest statement in the “Confidential to Editor” section, and submit your "Accept" recommendation.

Reviewer #1: All comments have been addressed

Reviewer #2: All comments have been addressed

2. Is the manuscript technically sound, and do the data support the conclusions?

Reviewer #1: Yes

Reviewer #2: Yes

3. Has the statistical analysis been performed appropriately and rigorously? 

Reviewer #1: Yes

Reviewer #2: Yes

4. Have the authors made all data underlying the findings in their manuscript fully available?

Reviewer #1: Yes

Reviewer #2: Yes

5. Is the manuscript presented in an intelligible fashion and written in standard English?

Reviewer #1: Yes

Reviewer #2: Yes

6. Review Comments to the Author

Reviewer #1: (No Response)

Reviewer #2: The revision has addressed all my concerns and comments well. I do not have any more comments on it. In my opinion I suggest to accept it.

7. PLOS authors have the option to publish the peer review history of their article (what does this mean?). If published, this will include your full peer review and any attached files.

Reviewer #1: No

Reviewer #2: No

---

## [Editor Report · Acceptance letter]

23 Aug 2021

PONE-D-21-14500R1 

Analysis and optimization of 15-minute community life circle based on supply and demand matching: A case study of Shanghai 

Dear Dr. Wang:

I'm pleased to inform you that your manuscript has been deemed suitable for publication in PLOS ONE. Congratulations! Your manuscript is now with our production department. 

Kind regards, 

on behalf of

Dr. Feng Chen 

Academic Editor

PLOS ONE